# Assessing the impact of introductory programming workshops on the computational reproducibility of biomedical workflows

**Ariel Deardorff** *

UCSF Library, University of California, San Francisco, California, United States of America

* ariel.deardorff@ucsf.edu

## Abstract

### Introduction

As biomedical research becomes more data-intensive, computational reproducibility is a growing area of importance. Unfortunately, many biomedical researchers have not received formal computational training and often struggle to produce results that can be reproduced using the same data, code, and methods. Programming workshops can be a tool to teach new computational methods, but it is not always clear whether researchers are able to use their new skills to make their work more computationally reproducible.

### Methods

This mixed methods study consisted of in-depth interviews with 14 biomedical researchers before and after participation in an introductory programming workshop. During the interviews, participants described their research workflows and responded to a quantitative checklist measuring reproducible behaviors. The interview data was analyzed using a thematic analysis approach, and the pre and post workshop checklist scores were compared to assess the impact of the workshop on the computational reproducibility of the researchers' workflows.

### Results

Pre and post scores on a checklist of reproducible behaviors did not change in a statistically significant manner. The qualitative interviews revealed that several participants had made small changes to their workflows including switching to open source programming languages for their data cleaning, analysis, and visualization. Overall many of the participants indicated higher levels of programming literacy, and an interest in further training. Factors that enabled change included supportive environments and an immediate research need, while barriers included collaborators that were resistant to new tools, and a lack of time.

**Data Availability Statement:** All data and accompanying metadata files are available from the Dryad repository at https://doi.org/10.7272/Q6RV0KW6

**Funding:** The author received no specific funding for this work.

**Competing interests:** I have read the journal's policy and the author of this manuscript has the following competing interests: the author is the co-chair of the Library Carpentry Advisory Group, one of the advisory groups of the Carpentries organization. Since completing this research, the author has taken on a project expanding outreach for the Carpentries in California academic libraries.

## Conclusion

While none of the workshop participants completely changed their workflows, many of them did incorporate new practices, tools, or methods that helped make their work more reproducible and transparent to other researchers. This indicates that programming workshops now offered by libraries and other organizations contribute to computational reproducibility training for researchers.

## Introduction

Scientific reproducibility–the idea that a scientific finding can be reproduced by others–has been called the "supreme court of the scientific system" [1]. However, there is a growing body of evidence that much of the research currently produced in academia cannot be reproduced. In a 2005 article, "Why Most Published Research Findings are False," John Ioannidis argued that most research results were likely false due to incorrect study design and statistical power [2]. This article led to several large-scale reproducibility studies including the Psychology Reproducibility Project, which analyzed 100 psychology studies and found that a large portion of them had findings weaker than what was claimed [3]. Failure to reproduce the results of a scientific study is not limited to one scientific domain. A 2016 survey in *Nature* found that more than 70% of researchers have tried and failed to reproduce another scientist's study, and even more alarming, more than half had failed to reproduce their own research [4]. One of the challenges of addressing the "reproducibility crisis" is that there are many competing definitions of reproducibility, each with their own solution. This paper focuses on computational reproducibility, defined as "the ability of a second researcher to receive a set of files, including data, code, and documentation, and to recreate or recover the outputs of a research project, including figures, tables, and other key quantitative and qualitative results" [5]. Computational reproducibility is a growing focus within the biomedical sciences and this aspect of reproducibility was the target of a recent National Academies report [6].

Computational reproducibility is particularly important in the biomedical sciences as nearly all fields and subdisciplines have grown increasingly data-intensive. Biomedical researchers increasingly must apply many of the skills that historically have been those of computer scientists, whether their work involves processing complicated files from electronic health records, collecting large genomic datasets, or building models of complex cellular relationships [6]. As their work changes, researchers must learn new methods and tools. It is no longer possible to rely on manual workflows in Excel; instead, researchers must learn new programming-based workflows to process and analyze their data. When these new workflows rely on open source programming languages such as R and Python and incorporate version control and detailed documentation, this work can be far more computationally reproducible [7–9]. Unfortunately, scientific training in this area has not always kept pace and many biomedical scientists do not learn basic programming or computational best practices as part of their graduate training. These researchers often struggle to engage in emerging areas of research and must seek out additional courses and workshops in order to use these new techniques [10].

At the University of California, San Francisco (UCSF), the library has partnered with the non-profit organization The Carpentries to teach introductory programming workshops to biomedical researchers with the goal of improving programming literacy and computational reproducibility. These Software Carpentry workshops are two-day hands-on events that cover

basic computing skills including the programming languages R or Python, version control with Git, and scripting in Unix [11]. The Carpentries regularly assesses participant learning in the workshops and has conducted a long-term assessment measuring uptake of certain behaviors associated with computational reproducibility, including use of programming languages and version control. The April 2020 analysis of the overall long-term survey results revealed that after a workshop 66% of respondents had started using a programming language or the command line [12]. In addition, 57% of participants "agreed" or "strongly agreed" that they had been able to make their analysis more reproducible. While this data is encouraging, these measures are self-reported, and it is difficult to tell how participants are using their skills to improve the computational reproducibility of their work. Since partnering with the Carpentries in 2016, UCSF has taught 12 workshops to over 700 UCSF researchers. While demand for the workshops is always high, there is a lack of evidence regarding the extent to which participation in the workshops leads researchers to adopt better computational practices. The aim of this study was to assess the impact of introductory programming workshops on biomedical researchers' workflows. More specifically, the goal was to discover if participants would change their workflows to incorporate new tools and methods learned in the workshops and thereby make their research practices more computationally reproducible.

## Methods

To learn about the impact of programming workshops on researcher workflows, this study used a mixed methods approach, consisting of semi-structured in-depth interviews that included questions about the researchers' workflows as well as a quantitative checklist measuring reproducible behaviors. The author interviewed fourteen biomedical researchers about their workflows before they participated in a two-day introductory programming workshop, and again three months after they had completed the workshop. This study was approved by the University of California, San Francisco Human Research Protection Program Institutional Review Board (#18–25691). All participants provided written informed consent prior to participation. An analysis of a subset of the pre-workshop qualitative data as well as the methods below were reported in an earlier publication [13].

### Study recruitment

The author recruited fourteen UCSF researchers who registered for a two-day library-led introductory programming workshop in March 2019. These workshops covered an introduction to Git, Unix, and either R or Python (with approximately 36 registered for the R track and 36 registered for the Python track) and were open to anyone affiliated with the university. The number of participants was selected based on research indicating that 12 interviews is generally sufficient to gather most major themes in a qualitative study [14]. The inclusion criteria specified that participants must be currently involved in research and planning on staying at UCSF for 6 months (in order to reach them for follow-up interviews). These criteria left a total of 59 possible participants out of the 72 enrolled in the course. The author used stratified random sampling to select potential participants, and contacted 7 participants registered for the R workshop and 7 registered for the Python workshop. Of the initial 14 participants selected, 7 did not respond and 2 declined to participate. These were replaced by an additional 9 random participants until 14 were reached.

### The interviews

In January and February 2019, the author performed in-depth semi-structured interviews with participants before they took the programming workshop. Participants were asked to draw

their research process and describe the tools and methods they used, pain points in their workflows, and what they were hoping to learn in the workshop (workflow drawing template in S1 File, pre-workshop interview protocol in S2 File). At the end of the interview, the author administered a brief 6 question checklist of computationally reproducible practices (checklist in S1 Checklist). This checklist was compiled based on a literature review of recommended practices for computational reproducibility, as well as the Carpentries long-term survey [5,7,12,15–23]. The checklist questions asked whether as part of their workflows participants used programming languages like R, Python, or the command line for data acquisition, processing, or analysis; transformed step-by-step workflows into scripts or functions; used version control to manage code; used open source software; shared their code publicly; and shared their computational workflow or protocols publicly. Each question was given a score of 1 point for a total possible score of 6 points for a workflow that incorporated all of the elements.

In June of 2019, three months after they had completed the workshop, the author performed follow-up interviews with the original participants. The follow-up interviews focused on their thoughts on the workshop, changes they had made to their research workflows, plans for future changes, factors that enabled or prevented workflow changes, and any suggestions they had for improving future workshops (post-workshop interview protocol in S3 File). Participants responded once again to the reproducible checklist to see if there had been a measurable change.

The pre and post workshop interviews were conducted at either the UCSF Parnassus or Mission Bay campus and ranged from 20 to 45 minutes. Before the first interview, participants signed a consent form stating that they agreed to be recorded and that they understood that their anonymized data would be shared. The interviews were recorded and the audio was transcribed by the online service Rev.com. After transcription, the author read through each interview transcript while listening to the audio recordings to ensure the content was faithfully reproduced, addressing any errors (for example, "are" instead of "R") as they were discovered. Finally, the author redacted names of people and groups and generalized research topics to preserve anonymity.

## Data analysis

To analyze the quantitative data, the author converted the checklist totals to numeric scores and performed a two-tailed t-test to measure for statistically significant changes in behaviors before and after the workshop.

The qualitative data was analyzed using the applied thematic analysis framework–a methodology inspired by grounded theory, positivism, interpretivism, and phenomenology [24]. For the pre and post workshop interviews, the author used an inductive approach to read through the transcripts, identify major themes, and create corresponding structural and thematic codes (code book available in S1 Data). These codes were then elaborated in the codebook and applied using an iterative approach. All coding was performed using the online data analysis tool Dedoose.

## Results

### Participant demographics

The majority (9 of 14) of research participants were postdoctoral researchers, followed by three research staff, one graduate student, and one faculty member. These demographics were in line with the typical audience of a UCSF programming workshop. The departmental representation was also similar to a typical workshop with a larger group from neurology (3 of 14), developmental and stem cell biology (3 of 14), and immunology (2 of 14), and the balance

coming from orthopedic surgery, neuroscience, neurological surgery, anatomy, pharmacy, and bioethics. As the workshops were marketed to beginning programmers, 13 of 14 described themselves a "novice" or "beginner" programmer, and only one participant considered themselves to be an "intermediate" programmer. During the second round of interviews in June 2019, one of the original participants declined to participate as they had not attended both days of the workshop, and another did not respond to emails. Therefore, only 12 researchers participated in the post-workshop interviews.

## Checklist scores before and after the workshop

The average score for the pre-workshop checklist was 1.6 out of 6 and ranged from 0 to 3, indicating lower levels of applied best practices in computational reproducibility. Of the six questions, participants scored highest on the use of programming language like R, Python, or the command line at some point in their workflow (n = 7) and using an open source tool (n = 7) (see Table 1). Five participants said they shared their code publicly, three said they shared their computational workflows publicly, one person said they transformed step-by-step workflows into scripts, and none of the participants used version control.

Three months after completing the workshop, the average score increased from 1.6 to 2.2 and ranged from 0 to 6, however this was not a statistically significant difference (t(11) = -1.04, p = 0.318). The post-workshop scores revealed increases in using programming languages like R or Python in their work (n = 8), transforming step-by-step workflows into scripts (n = 2), using version control (n = 2), using open source tools (n = 10), and decreases in sharing their code publicly (n = 2), and sharing their computational workflows publicly (n = 2). The interviews revealed that these decreased scores were likely due to a better understanding of what it means to share code and computational workflows, rather than an actual change in workflows. Previously, participants had mostly considered sharing to mean exchanging code or protocols with others in their lab or team. After the workshop, participants had a better understanding of what public code sharing could look like (for example sharing code on GitHub or in a repository like Zenodo) and realized that this was not part of their current workflow.

## Changes in participant workflows

While their overall checklist scores didn't change much, many of the participants indicated in the qualitative interviews that they had changed their workflow to incorporate programming languages in new ways throughout, including switching the tools they used for data cleaning, visualization, and analysis. One participant reported that they started using R to clean their data instead of their previous manual data cleaning in Excel. Two of the Python workshop attendees started using Python to plot figures and visualize data (although one said that they still preferred R). For data analysis, two of the participants shared that they had switched from

**Table 1. Total checklist scores before and after the workshop.**

| Question | PreTest Total (n = 14) | PostTest Total (n = 12) |
|---|---|---|
| Use programming languages like R, Python, or the command line for data acquisition, processing, or analysis | 7 | 8 |
| Transform step-by-step workflows into scripts or functions | 1 | 2 |
| Use version control to manage code | 0 | 2 |
| Use open source software | 7 | 10 |
| Share your code publicly | 5 | 2 |
| Share your computational workflow or protocols publicly | 3 | 2 |

proprietary tools like SPSS and Stata to R, saying that attending the workshop had helped them make the case to their collaborators that R was just as powerful a tool. Finally, two of the participants who enrolled in the Python class with an R background decided that R would actually be adequate for their work, with one stating that "actually my Python workshop has convinced me to go to quit Python. . ."

Participants had also made changes to make their workflows more programmatic, using the command line to download large datasets or using GitHub to find and share code. Four of the researchers reported that they started using Unix at some point to search and retrieve data files in their computers or download large datasets. One shared that "I didn't know anything about Terminal before so now I know what that is and how to basically use that. I think that has helped." Three participants stated that they had looked for code on GitHub and one of the more computationally advanced researchers had changed their workflow to incorporate sharing code with their collaborators on GitHub, quoting:

> "I'm using GitHub as well to share code with the people I work, on the team I am working. So it's also an interesting thing because for instance, some of the things that they do is I code something in R. I may build R shiny, I can use an interface for someone that is not fluent in R that can use whatever process. And, and now you can host that in the GitHub and they can directly just run one line of code and run that differently from GitHub and that's very useful. So to share code and so forth. So I've been using that as well."

## Plans for future workflow changes

While not all the researchers had been able to make changes in the three months since the workshop, many of them had ideas for ways they would change their workflows in the future. At least half of participants shared that they wanted to use R or Python for their data analysis going forward. Specific plans included using R to characterize cell types for sequencing data, analyzing images in Python, and trying Python to analyze single cell data. For some researchers, this meant taking ownership of a step that had previously been done by a collaborator. Describing their relationship with their team's bioinformaticist, one said:

> "So, I think, I don't know if I'll be doing all of it, but at least more together. Really, I was handing everything off to her. She'd do everything on her computer and then I'd only see figures weeks later. And so this would actually be like handling the data myself, doing some in R, if I can't, or am having issues like helping her, helping me kind of troubleshoot those things. Or even if she eventually does do some of the analysis, I'll know what she's done specifically."

Other researchers planned to switch to R or Python for their data visualization, indicating that they thought they could make better publication-quality figures. Two of the Python learners decided that they would eventually make the switch from R to Python for most of their analysis as they felt it was a simpler language that would have more applicability outside of academia. Finally, four of the participants expressed interest in using GitHub to share and version their scripts once they were writing more of their own code or had more autonomy. One researcher shared:

> "But in the future, if I were running my own projects, and I were more familiar, I would definitely try to use GitHub and all that because from what I saw, it's a good way to connect with other people on projects and I know people who use it as well. So just for project and

practice, it would have been good for a personal growth thing. But for research with my PI, yeah. She's not familiar, so I don't think we would use it."

### Increased programming literacy

While not every participant made changes to their workflow, the majority of them came out of the workshops with new insight into the language and fundamentals of programming. When asked about their biggest takeaway, several participants spoke about the fact that the workshop had helped de-mystify a complicated topic. One reported that they felt more comfortable talking about programming with their collaborators, and that they felt "like I could understand a little bit more what the more informatics people in my lab are doing day to day and then they talk about stuff and I'm like 'Oh, I know those words' like you sort of get to know the techniques that they're using a little bit more and the different software and stuff." Another felt they had started to see new ways to apply programming to their work. Another shared that learning Unix had helped them understand how their computer actually worked: "It's like being given a map of where you live suddenly and then, oh, there, that's where stuff is. So that was very useful."

### Interest in further training

A major takeaway from the study was that the workshop prompted many of the participants to seek more training opportunities. For some participants, the workshops helped clarify what they needed to learn. One person said, "I do feel like before it just all felt like very much a black box and now I at least kind of feel like I know what I need to learn." For others, the workshop helped them lay a foundation on which they wanted to build. One participant indicated that after learning the basics of R, "I'm not as anxious so I think that's also why I have been comfortable signing up for courses [that] are just like more data intensive. So that's definitely useful. And I want to attend more of those." Almost all participants said that they had explored further training and many had already taken another workshop. Three participants mentioned a specific single-cell sequencing data analysis workshop offered at UC Davis, another said they had registered for an Ed-X data science course, and a third said they were looking into the UC Berkeley extension courses.

### Enablers and barriers to change

When asked what helped them implement the workshop content into their workflows, the participants cited a number of factors. Some participants reported being able to quickly apply their new programming skills because they had an immediate research need. One postdoctoral researcher shared that they wanted to analyze RNA sequencing data and that they were the only one on their team who had the necessary programming skills. Others were able to practice on relevant datasets and scripts and see how they might use it with their own work. One of the most commonly cited factors was a supportive research environment; examples included a principal investigator (PI) who didn't know programming but saw the benefit for the lab, and a group of colleagues who already knew programming. One person shared, "I think it's both support from people in the lab. Seeing postdocs and graduate students who either have learned this and are really good or are actively learning it right now and working toward these goals. Made me be like, 'Oh, I would really like to have that skill.' I think that was really vital."

The participants mentioned many different barriers to effectively incorporating more programming into their workflows. For some, it was because their PI/collaborators were resistant to new tools or needed to be convinced to adopt new methods. In one instance, the researcher

recounted that they were unable to use GitHub because their PI preferred email. Another shared that their PI "doesn't have the knowledge of this, so he probably doesn't know how useful it can be, and it doesn't matter for them, they just need the results. If you visualize the data nicely, that's enough for them." Several participants also shared that they hadn't been able to implement new procedures because they didn't want to make changes in the middle of a project or had been stuck in the wet lab stage of their project. Overall the biggest barrier to change was a lack of time. While several participants saw the benefit of programming, they were not able to make time to learn enough to implement the new tools. One researcher summed up this problem:

> "So after, right after attending the workshop, I started using Python. But the thing is obviously I had to fix some problems. So, because I had the, I knew the basics, but then when I needed to do my own things and ask specific questions, I was not able to. And so, in order to be quick and to get my things done without wasting too much time . . .. I just continued with my usual way of proceeding."

## Discussion

The researchers in this study did not register for the programming workshops because they wanted their work to be more computationally reproducible; they registered so they could understand the work of their collaborators, work with new kinds of research, or analyze their data independently [13]. Many of the participants, however, did come away from the workshops with new practices, tools, or methods that made their work more reproducible and transparent to other researchers. Some participants switched from manual data cleaning in Excel to scripted data cleaning in R. Others moved from expensive proprietary software to open source tools. Some realized that they could share their code and protocols outside their labs. Finally, some started versioning their code to better track their work. While very few of the participants completely changed their workflows or adopted all the techniques after attending the workshop, all of them learned or tried something new. As with many aspects of reproducibility, computational reproducibility is often achieved through gradually adopting better practices, and all the participants were able to make some slight adjustment to their practices.

The results of this study are similar to many of the findings of the Carpentries' own long-term survey which included respondents from a wide variety of disciplines and career stages. Like the Carpentries' survey revealed, most of the UCSF researchers' implemented workflow changes involved integrating R and Python in new ways (66% of Carpentries respondents) rather than using version control to manage code (31%), or transforming step-by-step workflows into scripts (25%) [12]. The fact that many of the UCSF researchers' comments revealed an increase in programming literacy can be compared with the 79% of Carpentries respondents who indicated more confidence using the tools taught in the workshop. Finally, like the UCSF researchers, respondents to the Carpentries survey were also highly motivated to seek further instruction, with 89% reporting that they "agreed" or "strongly agreed" that they were motivated to seek more knowledge. As the Carpentries survey does not ask about what might have helped or hindered participants from implementing new skills, it may be that those identified in this study–including the enabling factors of having an immediate research need and the barriers of lack of time—also apply to researchers outside the biomedical sciences.

Beyond concrete workflow changes, one of the major outcomes of the workshop was enabling the biomedical research participants to start using the tools and approaches of computer scientists. Many of them expressed new understanding of how their computers work,

how to write a useable script, or what a scripted analysis could look like. They shared moments of revelation when finally understanding the language of their collaborators, or getting a glimpse of new ways of working. As biomedical research becomes increasingly data intensive, it is important that biomedical researchers–many of whom never learned to program—see tools like programming as attainable and essential parts of their workflow. While the participants in this study appear to have started down that path, they also struggled to make the switch to fully integrate programming, citing lack of time to truly learn new practices. Wrapped up in these statements was the idea that the "science" always took first priority and that learning to program was less essential, a potentially dangerous assumption that could prevent biomedical researchers from making full use of the tools available to them.

While some biomedical graduate programs are teaching programming as part of the curriculum, there is still a need to train the current postdoctoral researchers, research staff, and faculty who need these crucial skills in order to continue excelling in their areas of research. Introductory programming workshops like the ones offered by the Carpentries can be an excellent way for libraries and other organizations to jumpstart this learning process, but organizers and attendees should keep in mind that one workshop cannot teach all the skills necessary, and learners might need ongoing support to make the switch to new practices. Ideally, an introductory programming workshop should give researchers a taste of programming basics and possibilities and point them in the direction of further learning. It will likely be a gradual process, but any new practices can help make research more reproducible.

## Limitations

The results of this study are drawn from a small cohort of biomedical researchers at UCSF.

The data are representative of UCSF Carpentries workshops, but given the overrepresentation of postdoctoral researchers they might not reflect the experience of other biomedical researchers or researchers outside of the health sciences. While the participants in this study had similar demographic characteristics to the workshops overall (in terms of roles and departments), it is possible that the recruited participants who declined to participate or did not respond to the research invitation might have had different goals or expectations than those who did. The analysis and coding for this project was also performed solely by the author, and a different researcher might have interpreted slightly different themes. Finally, because of the small sample size, the quantitative analysis lacked appropriate power. A more well-defined checklist administered to a larger group might have revealed richer, more accurate data.

## Conclusion

Introductory programming workshops can be an excellent way for libraries and other organizations to contribute to biomedical research reproducibility. While none of the participants in this study completely transformed their workflows, many adopted new tools and practices including switching from proprietary to open source tool, incorporating more programmatic elements, and versioning their code. Participants also gained new insight into the fundamentals of programming and ideas and plans for changing their workflows in the future. These results indicate that researchers who learn new computational skills are able to improve their workflows, produce more transparent results, and contribute to more reproducible science.

## Supporting information

**S1 Checklist. Reproducibility checklist.**
(PDF)

**S1 File. Workflow drawing template.**
(PDF)

**S2 File. Pre-workshop interview protocol.**
(PDF)

**S3 File. Post-workshop interview protocol.**
(PDF)

**S1 Data. Qualitative analysis codebook.**
(CSV)

## Acknowledgments

The author would like to thank Kristine R. Brancolini, Marie Kennedy, and the Institute for Research Design in Librarianship (IRDL) for guidance on this project. Additional thanks to Savannah Kelly, Jill Barr-Walker, and Catherine Nancarrow for feedback on this manuscript.

## Author Contributions

**Conceptualization:** Ariel Deardorff.

**Data curation:** Ariel Deardorff.

**Formal analysis:** Ariel Deardorff.

**Investigation:** Ariel Deardorff.

**Methodology:** Ariel Deardorff.

**Project administration:** Ariel Deardorff.

**Validation:** Ariel Deardorff.

**Writing – original draft:** Ariel Deardorff.

**Writing – review & editing:** Ariel Deardorff.

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
