## [Decision Letter · Decision Letter 0]

28 Apr 2020

PONE-D-20-06264

Assessing the impact of introductory programming workshops on the computational reproducibility of biomedical workflows

PLOS ONE

Dear Ms Deardorff,

Thank you for submitting your manuscript to PLOS ONE. After careful consideration, we feel that it has merit but does not fully meet PLOS ONE’s publication criteria as it currently stands. Therefore, we invite you to submit a revised version of the manuscript that addresses the points raised during the review process.

In particular, even though both expert reviewers find the work interesting, they highlight the fact that the data used for the analysis is quite limited. Please, consider using data already available from the Software Carpentry impact assessment programme; if this is not possible (due to the different goals of your study) at least try to discuss/compare your findings in the context of the more generic assessments available therein. I would expect that a revised version of the manuscript will carefully address all points raised by the reviewers, which I believe will strengthen the main message of your work.

We would appreciate receiving your revised manuscript by Jun 12 2020 11:59PM. To enhance the reproducibility of your results, we recommend that if applicable you deposit your laboratory protocols in protocols.io, where a protocol can be assigned its own identifier (DOI) such that it can be cited independently in the future. For instructions see: http://journals.plos.org/plosone/s/submission-guidelines#loc-laboratory-protocols

We look forward to receiving your revised manuscript.

Kind regards,

Vasilis J Promponas

Academic Editor

PLOS ONE

2. Please provide additional details regarding participant consent. In the ethics statement in the Methods and online submission information, please ensure that you have specified whether consent was informed.

"I have read the journal's policy and the author of this manuscript has the following competing interests: the author is the co-chair of the Library Carpentry Advisory Group, one of the advisory groups of the Carpentries organization. Since completing this research, the author has taken on a project expanding outreach for the Carpentries in California academic libraries."

"The author would like to thank the Institute for Research Design in Librarianship (IRDL) for support with this project. IRDL is partially funded by the Institute of Museum and Library Services grant RE-40-16- 0120-16."

"The author received no specific funding for this work."

Reviewers' comments:

Reviewer's Responses to Questions

**Comments to the Author**

1. Is the manuscript technically sound, and do the data support the conclusions?

Reviewer #1: Yes

Reviewer #2: Yes

2. Has the statistical analysis been performed appropriately and rigorously? 

Reviewer #1: Yes

Reviewer #2: Yes

3. Have the authors made all data underlying the findings in their manuscript fully available?

Reviewer #1: Yes

Reviewer #2: Yes

4. Is the manuscript presented in an intelligible fashion and written in standard English?

Reviewer #1: Yes

Reviewer #2: Yes

5. Review Comments to the Author

Reviewer #1: The work presented here focuses on the impact that computational training workshops can have to the reproducibility practices employed by researchers in Life Sciences. In particular, the work describes a study performed by surveying the practices of biomedical researchers at UCSF before and after their participation to an introductory programming workshop. The survey itself was performed through short interviews (20-40 minutes long), with a randomized selection of 14 participants from within a set of 52 relevant cases (i.e. people involved in research, and planning on staying at UCSF for 6 months at least). The interviews were transcribed and both quantitative and qualitative data were analyzed using the online data analysis tool Dedoose. The overall analysis showed that there were notable (but not necessarily statistically significant) changes to the participants' research practices. These include a transition from commercial to open source tools, increased programming literacy and a definite interest for further training.

Overall the paper addresses an open question within the training communities, and especially within Life Sciences, which is to quantify and assess the impact of participation to a computational workshop regarding the practices employed by researchers. The work reflects similar studies being done by initiatives such as ELIXIR (within Europe) and the Carpentries (at a more global level), with the additional value of including a short interview aspect to the more standardized surveys. Overall it's an interesting work, but there are a few points that may be worth re-assessing

1. A main concern is about the overall impact/power of this approach, especially keeping the following two points in mind:

- The survey was performed on learners that participated in Software Carpentry workshops (although this point is not specifically mentioned in the text, I believe that this is an implied statement, also supported by the description of the workshops - page 6, line 125).

- The Carpentries maintain a fairly rigorous impact assessment programme, with a number of reports and analyses available on their website (https://carpentries.org/assessment/) under Zenodo DOIs, and including an "Analysis of Software and Data Carpentry’s Pre- and Post-Workshop Surveys" and a fairly recent "Analysis of The Carpentries Long-Term Surveys" (April 2020)

Although the work present here is much more detailed and focused on the reproducibility aspects (as opposed to the more streamlined and "templated" analysis that the Carpentries offer), at the same time the main driving point do share a significant overlap. Given that, due to the small sample size, the quantitative analysis lacks appropriate power, as also highlighted within the manuscript, it may be worth investigating how the interview data could be complemented by data gathered through the Carpentries impact assessment programme. In particular, this would hopefully underline the insights gained through this work, and connect to a larger number of research communities that are transitioning to more computational approaches (such as the Social Sciences, similarly to Life Sciences)

2. Another point is with regards to the existing bias due to the academic background of the learners. Although there is a representation of several different levels within the cohort, there is a significant bias in the "postdoctoral researcher" level with significantly less participation from other levels. Although this is not surprising nor unexpected, it would also make sense to include a stratified analysis of the results as it would be interesting to see the differences in impact from the workshop at different levels of academia.

3. A final point to highlight is that the actual analysis of the data (i.e. the computational notebook and/or the script) is not available. It is understood that the Dedoose platform was used for analyzing such data; however, I believe that, if the specific computational steps were available, it would greatly increase the further re-use of both data and insights gained through this work.

Reviewer #2: This is an interesting study. We all know that it is good for biomedical researchers to have some computing knowledge in order to make their research reproducible and make the right choice in terms of tools (and libraries), but I have not come across many studies that actually talk about it.

Initially I was a bit worried about the sample and it's small size, in terms of statistical power, but then the authors have actually mentioned about them as limitations. So, readers will be aware of this when they read this. I checked all the forms etc uploaded in DRYAD.

6. PLOS authors have the option to publish the peer review history of their article (what does this mean?). If published, this will include your full peer review and any attached files.

Reviewer #1: Yes: Fotis E. Psomopoulos

Reviewer #2: Yes: Shakuntala Baichoo

---

## [Author Response · Author response to Decision Letter 0]

18 May 2020

I have detailed my response to reviewer comments in the file called "Response to Reviewer Comments." In brief, I added more information from the Carpentries long-term survey to compare the results of my study with that data. I have also published additional stages of my analysis process in Dryad to make the process more transparent.

---

## [Decision Letter · Decision Letter 1]

23 Jun 2020

Assessing the impact of introductory programming workshops on the computational reproducibility of biomedical workflows

PONE-D-20-06264R1

Dear Dr. Deardorff,

We’re pleased to inform you that your manuscript has been judged scientifically suitable for publication and will be formally accepted for publication once it meets all outstanding technical requirements.

Kind regards,

Vasilis J Promponas

Academic Editor

PLOS ONE

Additional Editor Comments (optional):

I would like to bring to your attention that Plos One does not copyedit accepted manuscripts. Therefore, you should make sure to thoroughly check your manuscript for any remaining typos etc.

One such case is at "Line 56" where there is a period missing in the end of the sentence.

Reviewers' comments:

Reviewer's Responses to Questions

**Comments to the Author**

1. If the authors have adequately addressed your comments raised in a previous round of review and you feel that this manuscript is now acceptable for publication, you may indicate that here to bypass the “Comments to the Author” section, enter your conflict of interest statement in the “Confidential to Editor” section, and submit your "Accept" recommendation.

Reviewer #1: All comments have been addressed

Reviewer #2: All comments have been addressed

2. Is the manuscript technically sound, and do the data support the conclusions?

Reviewer #1: Yes

Reviewer #2: Yes

3. Has the statistical analysis been performed appropriately and rigorously? 

Reviewer #1: Yes

Reviewer #2: Yes

4. Have the authors made all data underlying the findings in their manuscript fully available?

Reviewer #1: Yes

Reviewer #2: Yes

5. Is the manuscript presented in an intelligible fashion and written in standard English?

Reviewer #1: Yes

Reviewer #2: Yes

6. Review Comments to the Author

Reviewer #1: (No Response)

Reviewer #2: Author has addressed all comments, especially w.r.t. Software Carpentry workshops' info and has worked on the list of references.

7. PLOS authors have the option to publish the peer review history of their article (what does this mean?). If published, this will include your full peer review and any attached files.

Reviewer #1: Yes: Fotis Psomopoulos

Reviewer #2: No

---

## [Editor Report · Acceptance letter]

26 Jun 2020

PONE-D-20-06264R1 

Assessing the impact of introductory programming workshops on the computational reproducibility of biomedical workflows 

Dear Dr. Deardorff:

I'm pleased to inform you that your manuscript has been deemed suitable for publication in PLOS ONE. Congratulations! Your manuscript is now with our production department. 

Kind regards, 

on behalf of

Dr. Vasilis J Promponas 

Academic Editor

PLOS ONE